# Tumor-Associated Lymphatics Upregulate MHC-II to Suppress Tumor-Infiltrating Lymphocytes

**DOI:** 10.3390/ijms232113470

**Published:** 2022-11-03

**Authors:** Claire Y. Li, Hyeung Ju Park, Jinyeon Shin, Jung Eun Baik, Babak J. Mehrara, Raghu P. Kataru

**Affiliations:** The Department of Surgery, Division of Plastic and Reconstructive Surgery, Memorial Sloan Kettering Cancer Center, New York, NY 10065, USA

**Keywords:** lymphatic immunomodulation, tumor microenvironment, tumor-infiltrating lymphocytes

## Abstract

Steady-state lymphatic endothelial cells (LECs) can induce peripheral tolerance by presenting endogenous antigens on MHC class I (MHC-I) molecules. Recent evidence suggests that lymph node LECs can cross-present tumor antigens on MHC-I to suppress tumor-specific CD8+ T cells. Whether LECs can act as immunosuppressive cells in an MHC-II dependent manner in the local tumor microenvironment (TME) is not well characterized. Using murine heterotopic and spontaneous tumor models, we show that LECs in the TME increase MHC-II expression in the context of increased co-inhibitory signals. We provide evidence that tumor lymphatics in human melanoma and breast cancer also upregulate MHC-II compared to normal tissue lymphatics. In transgenic mice that lack LEC-specific MHC-II expression, heterotopic tumor growth is attenuated, which is associated with increased numbers of tumor-specific CD8+ and effector CD4+ T cells, as well as decreased numbers of T regulatory CD4+ cells in the TME. Mechanistically, we show that murine and human dermal LECs can take up tumor antigens in vitro. Antigen-loaded LECs in vitro can induce antigen-specific proliferation of CD8+ T cells but not CD4+ T cells; however, these proliferated CD8+ T cells have reduced effector function in the presence of antigen-loaded LECs. Taken together, our study suggests LECs can act as immunosuppressive cells in the TME in an MHC-II dependent manner. Whether this is a result of direct tumor antigen presentation on MHC-II requires additional investigation.

## 1. Introduction

Tumors develop in complex tumor microenvironments (TME) consisting of stromal cells, immune cells, and extracellular matrix components; dynamic crosstalk between these components influence tumor progression and metastasis [1,2,3]. Among the various soluble immunosuppressive factors, cell-surface molecules, and cell types so far identified in the TME as potential therapeutic targets, lymphatics have only recently emerged as a potential candidate [3,4,5]. Lymphatics in the TME have been largely characterized for their role in tumor dissemination and transport of tumor antigens and immune cells [6,7,8]. However, there is emerging evidence that lymphatics may also play direct immunomodulatory roles in the TME [9,10]. This hypothesis is based on studies showing that physiologic lymphatic endothelial cells (LECs) are active immunomodulatory cells that can present antigens in the context of major histocompatibility complex class I (MHC-I) and possibly class II (MHC-II) molecules [11,12,13,14,15,16,17]. For instance, steady-state lymph node (LN) LECs can present peripheral tissue self-antigens on MHC-I to induce CD8+ T cell tolerance [13,17]. Steady-state LN LECs can also promote cellular apoptosis by cross-presenting exogenous antigens to CD8+ T cells [18]. The constitutive expression of co-inhibitory molecules such as PD-L1 and the lack of co-stimulatory molecules play a key role in LEC-induced CD8+ T cell tolerance [17,19,20].

However, the ability of LECs to directly present antigens on MHC-II molecules to affect T cell fate is still controversial. While one study suggests that LEC cannot present directly to CD4+ T cells due to the absence of MHC-II peptide loading machinery, others have shown that LECs can indeed present to CD4+ T cells leading to the maintenance of CD4+ T regulatory (Treg) populations [16,21,22]. Furthermore, because these studies were conducted under steady-state conditions, the capability of LECs to act as antigen presenting cells (APCs) in the tumor setting has not been extensively investigated. One study that did explore the role of LECs in the tumor setting showed that LECs in tumor-draining LNs can cross-present tumor antigens on MHC-I to induce tumor-specific CD8+ T cell apoptosis [23]. Additionally, recent evidence suggests that soluble factors in the TME can also induce an immunosuppressive phenotype in tumor-associated lymphatics [10,24]. However, in the context of immunotherapy, VEGF-C-induced lymphangiogenesis were shown to potentiate immunotherapy in murine tumor models [25]. Given these inconsistent roles of tumor lymphatics, the role and mechanism of local lymphatic immunomodulation in the TME requires additional investigation.

Here, we demonstrate that LECs in the TME upregulate expression of MHC-II in both murine tumor models and in human melanoma and breast cancer specimens. In mice, upregulation of antigen presenting machinery occurs in the context of increased co-inhibitory molecules and is dependent on IFNγ. By using an inducible, lymphatic-specific MHC-II knockout mouse model, we show that abrogation of LEC MHC-II expression results in attenuated tumor growth which is associated with increased tumor-specific and effector TILs in the TME. Mechanistically, we demonstrate that LECs can take up tumor antigens and induce antigen-specific CD8+ T cell proliferation with reduced effector function. Notably, our results suggest a tolerogenic role for LECs locally in the TME and implicate MHC-II specifically in this pathway. Our study expands on those recently published by Gkountidi et al. using a different mouse model of lymphatic specific MHC-II deletion and provides novel evidence of the immunosuppressive potential of LECs in human tumors [26].

## 2. Results

### 2.1. Peritumoral Lymphatics Upregulate MHC-II Expression in the Context of Co-Inhibitory Molecules

LN LECs are transcriptionally distinct from peripheral tissue LECs and display increased expression of antigen presentation genes [27]. These characteristics suggest that LN LECs have immunomodulatory roles distinct from their peripheral counterparts [27]. Consistent with previous findings, we noted that LN LECs have significantly increased expression of MHC class II and co-inhibitory signal PD-L1 compared to ear skin LECs (Appendix A).

To test the hypothesis that the expression patterns of MHC-II and PD-L1 are modulated in the lymphatic vessels of the TME, we heterotopically implanted B16F10 tumor cells in the flanks of C57BL/6 wild-type (WT) mice and analyzed cell surface expression using flow cytometry (Figure 1A). Expression of MHC-II was increased in peritumoral dermal LECs compared with normal ear skin dermal LECs as early as 7 days following tumor implantation (Figure 1B). Increased MHC-II expression in peritumoral LECs persisted at later time points as well (Figure 1B and Appendix A). Peritumoral LEC PD-L1 expression was not changed in the early time points (days 7 and 14) we examined; however, at 21 days following tumor implantation, we noted a significant increase in PD-L1 expression in peritumoral LECs as compared to ear skin LECs (Figure 1B). Peritumoral LEC MHC-II expression was also significantly increased in mice 14 days after heterotopic tumor implantation with murine breast cancer cell line E0771, suggesting that upregulation of MHC-II expression in peritumoral LECs is independent of tumor type (Figure 1C and Appendix A). We saw a trend toward higher expression of PD-L1 in peritumoral LECs in E0771- bearing mice although this did not reach statistical significance at 2-weeks post tumor implantation similar to what was observed with B16F10- bearing mice (Figure 1C and Appendix A). To eliminate confounding effects of heterotopic tumor models, we repeated these experiments in a spontaneous tumor model by using MMTV-PyMT transgenic mice that develop mammary tumors. Similar to our observations in B16F10- and E0771-bearing mice, we found that peritumoral LECs in MMTV-PyMT mice had increased MHC-II and PD-L1 expression compared to ear skin LECs in the same animal (Figure 1D and Appendix A). At steady state, LECs do not express co-stimulatory molecules but does express high levels of inhibitory signals PD-L1, HVEM, and CD48; antigen presentation on LECs to cognate CD8+ T cells results in the expression of receptors on CD8+ T cells for PD-L1 and HVEM but not CD48 [19]. We asked if the increased expression of antigen presenting machinery in local peritumoral lymphatics also occurs in the context of increased co-inhibitory signals. Indeed, peritumoral LECs had increased expression of co-inhibitory molecules HVEM and CD48 compared to ear skin LECs; however, expression of PD-L2 was not significantly different (Figure 1E and Appendix A). This is consistent with published data showing low levels of LEC PD-L2 expression at steady state [19]. Immunohistochemical analysis of heterotopically implanted B16F10 tumors confirmed expression of MHC-II only in peritumoral LECs--lymphatics adjacent to the tumor margin and not in lymphatics by the skin margin--suggesting soluble factors in the TME are responsible for the upregulation of MHC-II in peritumoral LECs (Figure 1F).

We next asked if increased expression of LEC MHC-II in the TME also extended to human cancers. To answer this question, we analyzed human tissue microarrays of melanoma and breast tumor specimens and found higher expression of HLA-DR/DP/DQ in tumor-associated lymphatics compared normal skin and breast lymphatics (Figure 2). Publicly available transcriptomics data comparing human breast cancer LECs and normal breast tissue LECs further confirmed significantly increased expression of various MHC class I and II molecules in cancer-associated LECs (Appendix A). This data also demonstrated significantly increased expression of co-inhibitory molecule TNFRSF14 (HVEM) in peritumoral breast cancer LECs compared to normal breast LECs, consistent with our findings in B16F10- bearing mice (Appendix A). However, other co-inhibitory molecules CD48, PD-L1, and PD-L2 were not significantly different between breast cancer and normal breast tissue LECs (Appendix A). Taken together, these data show that peritumoral LECs in both animal and human tumors have increased expression of MHC-II compared to naïve peripheral LECs, suggesting local immunomodulatory roles for peritumoral lymphatics in human cancers.

### 2.2. IFNγ Induces Upregulation of MHC-II in LECs

The expression of steady-state MHC-II in LN LECs is dependent on both the IFNγ-inducible pIV of CIITA as well as DC-acquired MHC-II molecules [12,28]. We asked if increased MHC-II expression in peritumoral LECs is dependent on IFNγ or other immunologically active mediators found in abundance in the TME [29,30,31,32]. To answer this question, we treated human dermal LECs (hdLECs) with IFNγ, VEGF-C, IL-2, or TNF-α and found that IFNγ significantly increased expression of HLA-DRA and PD-L1 compared to no treatment, VEGF-C, IL-2 or TNF-α treatments (Figure 3A–D). These results were confirmed with flow cytometry (Figure 3E,F). IFNγ treatment also significantly increased expression of the MHC-II peptide loading molecule HLA-DM (Figure 3A). This is important because the lack of peripheral LECs HLA-DM expression was hypothesized in one study to explain why peripheral LECs do not present antigens on MHC-II at steady state [16]. Interestingly, IFNγ treatment significantly decreased LYVE-1 expression in LECs—this finding in the context of increased expression of immunomodulatory molecules suggests a decrease in lymphatic phenotype toward a more immunomodulatory phenotype as a result of IFNγ (Figure 3A). In contrast, treatment of cultured LECs with VEGF-C, IL-2, or TNF-α resulted in a trend of decreased expressions of HLA-DM, PD-L1, and LYVE-1 (Figure 3B–D).

To determine if peritumoral LECs in vivo depended on IFNγ, we treated WT mice implanted with B16F10 with intraperitoneal injections of anti-IFNγ neutralizing antibody or IgG isotype control and analyzed MHC-II and PD-L1 expression in peritumoral LECs at 14 days post tumor implantation. MHC-II expression in peritumoral LECs in animals treated with anti-IFNγ antibody was significantly decreased compared to animals treated with IgG isotype control (Figure 3G,H). However, there were no differences between LEC PD-L1 expression between animals treated with anti-IFNγ antibody or IgG isotype control. Taken together, these data show that increased expression of MHC-II in local peritumoral LECs is dependent on IFNγ.

### 2.3. Generation of a Conditional, Lymphatic-Specific MHC-II^−/−^ Mouse Model

We next sought to determine the biological significance of increased LEC MHC-II expression in the TME. To answer this question, we generated a conditional lymphatic-specific MHC-II^−/−^ mouse by crossing Flt4CreER^T2^ mice with H2-Ab1-floxed mice, henceforward referred to as LEC^MHC-II−/−^ (Figure 4A). Treatment with tamoxifen in LEC^MHC-II−/−^ mice induced deletion of MHC-II mRNA in FACS-sorted LN LECs but not in LN CD11c+ or CD11b+ immune cells (Figure 4B,C). We confirmed these findings in LN LECs with flow cytometry and additionally showed that PD-L1 expression is not modulated by tamoxifen treatment in LEC^MHC-II−/−^ mice (Figure 4D,E). Peritumoral LEC MHC-II expression was also significantly decreased in these animals by a mean of 64.5% compared to WT controls; however, PD-L1 expression was again unchanged (Figure 4F,G). Since Flt4 can also be expressed on blood endothelial cells (BECs) and macrophages, we analyzed MHC-II expression in LN BECs and tumor-associated macrophages (TAMs) in LEC^MHC-II−/−^ mice compared to controls and found no significant difference (Appendix A).

### 2.4. Attenuated Tumor Growth in Lymphatic-Specific MHC-II^−/−^ Background Is Associated with Effector Phenotypes in TILs

To elucidate the role of peritumoral LEC MHC-II expression on tumor immune responses, we implanted B16F10 and E0771 tumor cells lines in the flanks of tamoxifen-treated LEC^MHC-II−/−^ and control mice. LEC^MHC-II−/−^ animals had attenuated tumor growth of both B16F10 and E0771 compared to controls (Figure 4H–K). We next sought to investigate the mechanism of attenuated primary tumor growth in the context of LEC-specific MHC-II knockout. LN lymphangiogenesis and peripheral skin lymphatic structures were grossly identical between LEC^MHC-II−/−^ and littermate controls (Appendix A). Additionally, using hindlimb skin FITC painting by FITC isomer in combination with acetone and dibutyl phthalate, we found that DC migration to the draining inguinal LN was not affected by LEC-specific MHC-II knockout (Appendix A). There were also no differences in the number of peritumoral LECs in LEC^MHC-II−/−^ mice compared to controls (Appendix A). We also show that the baseline number of splenic monocytes (DCs and macrophages), leukocytes (CD8+ and CD4+ T cells), and Tregs were unchanged in LEC^MHC-II−/−^ mice treated with tamoxifen compared to controls (Appendix A).

Given that lymphatic structure, drainage, peritumoral lymphatic density, and baseline immune cells were unchanged in the setting of LEC-specific MHC-II deletion, we hypothesized that increased peritumoral LEC MHC-II expression tolerized the host adaptive immune response to tumor antigens in a manner that is similar to the physiologic role of LECs in maintaining peripheral tolerance. We reasoned that disruption in LEC MHC-II dependent antigen presentation would alleviate immunosuppressive pressures in the TME, thus allowing for better tumor control by the adaptive immune response. To test our hypothesis, LEC^MHC-II−/−^ mice and controls were implanted with B16F10 coupled to foreign peptide OVA (B16F10-OVA), and leukocytes were quantified in various locations, including the primary tumor (TILs), the tumor-draining lymph node, and the spleen. Overall, CD8+ T cells were not different between LEC^MHC-II−/−^ and controls in the primary tumors or tumor-draining LNs but were higher in spleens of LEC^MHC-II−/−^ mice (Figure 5A,B). However, we found a significantly greater number of tumor specific OVA+ CD8+ T cells in the primary tumors and spleens of LEC^MHC-II−/−^ mice compared to controls (Figure 5C,D). There were no differences in the number of OVA+ CD8+ T cells in the tumor-draining LNs (Figure 5C,D). We found no differences in intracellular granzyme expression in OVA+ CD8+ cells between tumor-bearing LEC^MHC-II−/−^ mice and controls in any location (Figure 5E,F).

Because MHC-II primarily interacts with the TCR of CD4+ T cells, we hypothesized its deletion would also affect CD4+ phenotypes. The number of CD4+ T cells in the primary tumors, tumor-draining LNs, and spleens of tumor-bearing LEC^MHC-II−/−^ mice were not different from controls (Figure 6A,B). However, the number of Tregs (CD4+ CD25+ FoxP3+) were significantly lower in the tumors of LEC^MHC-II−/−^ mice but not different in tumor-draining LNs or spleens (Figure 6C,D). Interestingly, non-regulatory CD4+ T cells (CD4+ CD25- FoxP3-) also had a greater proportion of granzyme B+ cells in the TME of LEC^MHC-II−/−^ mice compared to controls (Figure 6E,F). These data show that attenuated tumor growth in LEC-specific MHC-II knockout is associated with increased numbers of tumor-specific CD8+ and effector CD4+ T cells, as well as decreased numbers of T regulatory CD4+ cells in the TME. Representative histologic sections corroborate decreased number of CD4+FoxP3+ double staining cells found in the peritumoral region of LEC^MHC-II−/−^ compared to controls; these sections also show grossly higher density of granzyme B staining CD8+ cells in the peritumoral region of LEC^MHC-II−/−^mice compared to controls even though this result did not reach statistical significance using flow cytometric analysis (Appendix A).

### 2.5. Cultured LECs Take Up Tumor Debris and Induce Antigen-Specific CD8+ T Cell Proliferation with Reduced Effector Function

We next hypothesized that peritumoral LECs can present tumor antigens to tolerize the host adaptive immune response. To determine if dermal LECs can present antigens, we first sought to determine if these cells were capable of tumor antigen uptake. When human and murine dermal LECs are cultured in the presence of CFSE labeled B16F10 tumor debris, labeled tumor debris was observed inside the cytoplasm of both murine and human dermal LECs (Figure 7A). We next asked if dermal LECs can present antigens to lymphocytes in an antigen-specific manner. To answer this question, we pulsed murine dermal LECs in vitro with whole OVA protein and cocultured pulsed or unpulsed LECs with CFSE-labeled naïve CD8+ OT-I and CD4+ OT-II cells whose TCRs are specific for OVA. OVA-pulsed and unpulsed DCs served as positive and negative controls, respectively; OT cells grown alone served as a CFSE staining control. As expected, OVA-pulsed DCs induced significant proliferation of OT-I and OT-II cells compared to unpulsed DCs, indicating antigen-specific presentation and assay validity (Figure 7B,C, Appendix A). OVA-pulsed LECs did not induce significant proliferation of OT-II cells, nor was there proliferation in the presence of OVA-pulsed LECs and un-pulsed DCs (Figure 7B and Appendix A). In contrast, OVA-pulsed LECs did induce significant proliferation of OT-I cells in co-culture as well as in the presence of unpulsed DCs (Figure 7C and Appendix A).

Given this data, we next focused on potential differences in functionality in OT-I proliferated cells co-cultured with antigen-loaded DCs compared to antigen-loaded LECs. We repeated our in vitro experimental setup and quantified granzyme B positivity in the last generation (CFSE-) of OT-I proliferated cells. As expected, there were significantly greater granzyme B+ cells in proliferated OT-I cells cocultured with OVA-pulsed DCs than unpulsed DCs (Figure 7E). We found significantly lower granzyme B positivity when OT-I cells were cultured with antigen-loaded LECs, either in co-culture or tri-culture with un-pulsed DCs, compared to OT-I cells cocultured with OVA-pulsed DCs (Figure 7D,E). We next sought to investigate if OT-I proliferation under pulsed LEC and unpulsed DC triculture conditions were indeed due to OVA uptake by LECs. To answer this question, we pretreated both unpulsed and pulsed LECs with NH_4_Cl, which inhibits antigen processing at the level of endosomal-lysosomal fusion. Pretreatment with NH_4_Cl negated OT-I proliferation under pulsed LEC and unpulsed DC triculture conditions (Figure 7F). Similarly, when LECs and DCs were pulsed with OVA at 4 °C to inhibit antigen uptake, there was no significant increase in OT-I proliferation as seen under standard 37 °C culture conditions (Figure 7G). Overall, these data show that LECs can engulf tumor antigens and that antigen-loaded LECs can induce antigen-specific CD8+ T cell proliferation albeit with reduced effector function.

## 3. Discussion

Recent advances in our understanding of lymphatic biology reveal that many of the physiologic functions of the lymphatic system can be hijacked by tumor cells for their proliferation and dissemination [33,34,35]. For example, steady-state lymphatic vessels transport macromolecules, lipids, and immune cells; however, these same conduits also act as a conduit for tumor cell metastasis [33]. Studies demonstrating a link between tumor lymphangiogenesis and metastasis support the hypothesis that increased access to lymphatic capillaries increases the likelihood of tumor cell dissemination [36,37,38,39,40,41]. On the other hand, priming of the anti-tumor response occurs in the tumor-draining LN, and ineffective lymphatic transport in the form of dysfunctional peritumoral lymphatics hamper the generation of an effective adaptive immune response [6,7,42]. The role of lymphatics in generating an effective anti-tumor immune response may explain the beneficial effects of lymphangiogenesis on immunotherapy reported in several studies [25,43].

Whether tumor cells also co-opt the ability of LECs to act as APCs is largely unknown. LECs in the tumor-draining LN can cross-present tumor antigens on MHC-I to induce tolerance of tumor-specific CD8+ T cells similar to the ability of steady-state LECs to present endogenous self-antigens on MHC-I to suppress autoreactive CD8+ T cells [23]. Studies investigating the ability of LECs to act as MHC-II restricted APCs at steady state are conflicting, and until, recently there were no reports of LECs acting as MHC-II restricted APCs in the tumor setting. Recently, Gkountidi and colleagues reported similar findings as this study describing the role of tumor-associated LECs as APCs in an MHC-II dependent manner [26]. In the present study, we show that in the TME, dermal LECs upregulate the expression of MHC-II. Gkoutidi et al. reported similar findings although using a VEGF-C model of murine melanoma B16F10. Therefore, our results add to the findings published by Gkoutidi et al. by showing that increased antigen presenting machinery in tumor associated lymphatics is VEGF-C independent and likely a function of the inflammatory milieu of the TME. Additionally, we found that the upregulation of MHC-II in peritumoral LECs is associated with a concurrent increase in the expression of co-inhibitory molecules HVEM, CD48, and PD-L1. These results suggest that LECs in the TME increase the expression of antigen presenting machinery in the context of increased immunosuppressive signals. Notably, we found the upregulation of LEC MHC-II in the TME of heterotopic and spontaneous mouse tumor models also occurs in human melanoma and breast cancer specimens.

Increased peritumoral LEC MHC-II expression is partially due to IFNγ, as MHC-II expression was significantly increased in hdLECs after treatment with IFNγ, and anti-IFNγ neutralizing antibody decreased peritumoral LEC MHC-II expression compared to animals treated with IgG isotype control. Additionally, PD-L1 and MHC-II loading molecule HLA-DM were both significantly increased under IFNγ stimulation. These results are consistent with those obtained independently from Gkoutidi et al. [26]. Work by Rouhani and colleagues suggests steady-state LECs cannot present endogenous antigens on MHC-II due to lack of MHC-II antigen loading molecule H2-M (HLA-DM in humans) [16]; based on our data and that by Gkoutidi et al., there is reason to suggest that in pathologic settings LECs upregulate expression of H2-M to load antigens on MHC-II [26].

Using two different lymphatic specific cre-promoters (Prox-1 and Flt4, respectively) to induce MHC-II deletion in LECs, Gkoutidi et al. and our laboratory arrived at the complementary finding that deletion of MHC-II expression in LECs results in impaired primary tumor growth [26]. In our experimental assay, peritumoral LEC MHC-II expression was not completely eliminated after induction of Cre-*lox* system. This could be the result of technical limitations of the Cre-lox system, induction of MHC-II expression by soluble factors in the TME in cells without efficient knockout, or even the transfer of MHC-II from DCs which has been previously reported [12]. We hypothesize that in a model in which LEC MHC-II deletion could be induced post tumor implantation to negate TME-dependent MHC-II upregulation, then the control of tumor growth seen as a result of LEC MHC-II abrogation would be even greater than what we reported in the present study.

Additionally, we found that attenuated tumor growth in LEC MHC-II knockout background was associated with an overall improved tumor-specific immune response as characterized by increased numbers of tumor-specific CD8+ and effector CD4+ T cells, as well as decreased numbers of Tregs locally. Gkoutidi et al. reported similar findings regarding TILs in their LEC-specific MHC-II^−/−^ transgenic mice but only when the animals were implanted with high VEGF-C expressing B16F10; in their report, a trend toward decreased Tregs and increased IFNg+ CD8+ in LEC MHC-II^−/−^ background using parental B16F10 was seen but this was not statistically significant [26]. The fact we saw significant differences in these TILs could be due to methodology, including time of analysis post tumor implantation. Using BM chimeric FoxP3 labeled cells, Gkoutidi et al. were able to elegantly devise a method to isolate Tregs from LEC^MHC-II−/−^ mice; this confirmed that LEC MHC-II^−/−^ was associated with an increased suppressive phenotype locally [26]. Also consistent with what was reported by Gkoutidi et al., we did not observe any difference in CD4+ or CD8+ T cell number or effector function in tumor-draining LNs [26]. Therefore, although the effect of LEC MHC-II knockout is not limited to peritumoral LECs alone and also affects LN LECs, our findings suggest that soluble factors in the TME can induce immunosuppressive activity in peritumoral LECs with less significant effects on LECs in the draining LN. The fact that we did not find any differences in LN LEC MHC-II expression in tumor-draining versus contralateral nodes in tumor-bearing WT mice supports this hypothesis (data not shown).

Our mechanistic experiments involving antigen-loaded mouse dermal LECs and antigen-specific OT-I and -II cells suggests that LECs cannot directly present to naïve CD4+ T cells, as evident by lack of OT-II proliferation. The fact that we did not see CD4+ T cell proliferation with antigen-loaded LECs does not eliminate the possibility that LECs can modulate CD4+ T cell fate, either directly or indirectly. Because we used only naïve CD4+ T cells in our experiment, it is possible that LECs present in an MHC-II restricted manner only to a subset of CD4+ T cells. Nörder et al. reported similar findings in which human MHC-II+ LN LECs cultured with allogeneic CD4+ T cells failed to induce proliferation; however, the supernatants of these LECs impaired DC-induced allogenic CD4+ T proliferation [44]. It is possible that a similar phenomenon occurs in the tumor setting and in the background of LEC MHC-II^−/−^, in which antigen-loaded LECs modulates CD4+ T cells in the TME not through direct antigen presentation but through elaboration of soluble factors.

Interestingly, antigen-loaded LECs did induce proliferation of CD8+ T cells; however, these proliferated CD8+ T cell had reduced effector function compared to conditions in which DCs are the antigen presenting cell type. This further supports the theory that LECs induce an immunosuppressive phenotype in CD8+ T cells. Although we found significantly higher numbers of tumor-specific CD8+ TILs in the background of lymphatic MHC-II knockout, we did not see significant differences in granzyme B positivity in these cells in vivo, as suggested by our in vitro data. This discrepancy highlights two limitations of this study and areas for further investigation. One, LECs in the TME are likely to interact with effector or memory T cells, whereas naïve CD8+ T cells were used in in vitro experiments. Two, in our in vitro assay, LECs in culture have intact MHC-II expression. Additional experiments will be needed to investigate the ability of LECs to present to effector or memory lymphocytes in an antigen-specific, MHC-II dependent manner. Lastly, it would be interesting to investigate the responsiveness of LEC MHC-II deficient mice bearing tumors to immunotherapy. Recent work by Vokali et al., show that LECs can prime a small set of naïve CD8+ T cells under physiologic conditions into memory cells that then differentiate into effector cells upon memory recall [45]. We did not investigate differences in memory CD8+ T cells in our experiments but it is possible that we would see decreased numbers of memory CD8+ T cells in LEC MHC-II knockout background according to this recent work [45]. Accordingly, MHC-II deletion in LECs could result in decreased immune activation due to decreased memory CD8+ T cells despite initial improved tumor control.

In conclusion, our data reveal that lymphatics can act as immunomodulatory cells in the TME. Compared to steady-state LECs, peritumoral LECs increase MHC-II expression in the context of increased co-inhibitory signals; knockout of lymphatic MHC-II alleviates the immunosuppressive influence of LECs to tip the balance in favor of the host anti-tumor response. Our findings warrant additional investigations into the role of LEC MHC-II in the setting of human tumors and immunotherapy.

## 4. Materials and Methods

### 4.1. Mice

Male and female C57/BL/6J (wild-type) and FVB/N-Tg (MMTV-PyMT) mice were purchased from the Jackson Laboratory (Bar Harbor, ME, USA). FLT4-CreER^T2^H2-Ab1-floxed mice were generated by crossing H2-Ab1-floxed mice (B6.129X1-*H2-Ab1^tm1Koni^*/J Jackson Laboratory, Stock#013181) mice with Flt4CreER^T2^ mice (a gift from S. Ortega, Centro Nacional de Investigaciones Oncolόgicas; Madrid, Spain) generated as previously described [46]. The expression of both transgenes was confirmed by genotyping (Transnetyx, genotyping service from transnetyx), and double-homozygous mice (LEC^MHC-II−/−^) were backcrossed for six to seven generations to ensure consistency. Flt4-cre recombinase was activated using three doses of 100 mg tamoxifen (Sigma-Aldrich, Cat# T5648; Saint Louise, MO, USA) injected intraperitoneally every other day starting at 6–8 weeks of age. Age-matched C57BL/6J mice were used as controls. All experiments were performed 2 weeks after the first dose of tamoxifen injection. All mice were maintained in a pathogen-free, temperature- and light-controlled environment and provided with a normal chow diet and fresh water ad libitum. After each experiment, animals were euthanized by carbon dioxide asphyxiation as recommended by the American Veterinary Medical Association.

### 4.2. Tumor Models

Melanoma implantation into mouse flanks was performed using B16F10-mCherry (gifted by Dr. David Lyden, Weill Cornell Medicine; New York, NY, USA) or B16F10-OVA (gifted by Dr. Sasha Rudensky, MSKCC; New York, NY, USA). Breast cancer implantation into mouse flanks was performed using E0771 cells (gifted by Dr. Jacqueline Bromberg, MSKCC; New York, NY, USA). Cells were grown at 37 °C in 5% carbon dioxide in Dulbecco’s Modified Eagle Medium (DMEM; Gibco; Grand Island, NY, USA) supplemented with 10% fetal bovine serum, 100 U/mL penicillin, 100 μg/mL streptomycin, and 400 μM L-glutamine. Cells were passaged every 2-3 days and detained with 0.05% trypsin-ethylenediaminetetraacetic acid (Gibco, Cat#25200056). A total of 1.0 × 10^6^ cells in 100 μL DMEM were injected intradermally into the hair-trimmed lateral flanks of anesthetized mice. Tumor growth was measured by digital calipers every 7 days starting on day 7 after implantation. Mammary tumors of MMTV-PyMT mice were resected for analysis from female mice after becoming palpable. For treatment with anti-IFNγ antibody experiments, WT mice were injected with intraperitoneal IFNγ neutralizing antibody (BioXCell, Cat# BE0055; Lebanon, NH, USA) or IgG control (BioXCell, Cat# BE0088) every other day starting one day prior to B16F10 flank injection; analysis was performed 14 days after tumor implantation.

### 4.3. Histology and Immunohistochemistry

Histology and immunohistochemistry staining were performed using standard protocols. Briefly, tissues were fixed in 4% paraformaldehyde (Thermo Scientific, Cat#AAJ19943K2; Waltham, MA, USA) at 4 °C, embedded in Tissue-Plus optimal cutting temperature (OCT) compound (Fisher Scientific, Cat#23-730-625; Waltham, MA, USA), and sectioned at 5–7 μm. For immunohistochemistry, nonspecific binding was blocked with a 5% donkey or goat serum (Equitech-Bio, Cat#SD30; Kerrville, TX, USA) for one hour at room temperature. All tissue sections were then incubated at 4 °C with the appropriate primary antibodies overnight. The following primary anti-mouse antibodies were used: LYVE-1 (Abcam, Cat#14917; Waltham, MA, USA), MHC-II (Abcam, Cat#25333; ), PD-L1 (R&D, Cat#AF1019), CD4 (R&D, Cat#554; Minneapolis, MN, USA), CD8 (Abcam, Cat#22378), Granzyme B (Abcam, Cat#255598), FoxP3 (Abcam, Cat#22510). Sections or whole-mount tissue preparations were subsequently washed with PBS with Triton X-100 (Sigma-Alrich) and incubated with corresponding fluorescent-labeled secondary antibody conjugates for five hours followed by 4,6-diamidino-2-phenylindole (DAPI; Molecular Probes/Invitrogen, Cat#D4571; Waltham, MA, USA) for ten minutes before mounting with Mowiol (Sigma-Aldrich). All sections were scanned using a Mirax slide scanner (Zeiss; Munich, Germany). Confocal and whole-mount images were imaged using an SP-5 upright confocal microscope (Leica Microsystems, Wetzlar, Germany). Analysis was completed using Pannoramic Viewer (3D Histech; Budapest, Hungary).

For human tissues, tumor and benign tissue microarrays were purchased from US Biomax (ME2081, BC081120f). Arrays were deparaffinized with xylene, rehydrated, and antigen retrieved by pressure cooking for 15 min in citrate buffer. Slides were pre-incubated with 5% donkey serum for one hour at room temperature. Melanoma and breast cancer tissue microarrays were incubated with primary antibodies mouse anti-HLA DR/DP/DQ antibody (clone CR3/42, Abcam, Cat#7856) at 1/100 dilution and goat anti-LYVE1 antibody (R&D, Cat#AF2089) at 1/400 dilution for 1 h at room temperature followed by incubation with the corresponding fluorescent-labeled secondary antibody conjugates (Invitrogen) for three hours at room temperature. Final incubation in DAPI and mounting was performed as described above.

### 4.4. In Vitro Cytokine Treatment

Human dermal LECs (PromoCell, Cat#C-12217; Heidelberg, Germany) were grown to be 60–80% confluent in endothelial growth medium-MV-2 (PromoCell, Cat#C-22022) before treatment with cytokines and growth factors. Human recombinant IFNγ (PeproTech, Cat# 300-02; 100 ng/mL; East Windsor, NJ, USA), VEGF-C (PeproTech, Cat# 100-20CD; 100 ng/mL), TNF-α (PeproTech, Cat# 300-01A; 10 ng/mL), IL-2 (PeproTecc, Cat# 200-02; 10 ng/mL) or media only (control) was added to cultures for 24 h before downstream analysis.

### 4.5. Quantitative PCR Analysis

PCR analysis for HLA-DRA (Qiagen, GeneGlobe ID QT00089383; Germantown, MD, USA), PD-L1 (Qiagen, GeneGlobe ID QT00082775), HLA-DM (Qiagen, GeneGlobe ID QT00197288), and LYVE-1 (Qiagen, GeneGlobe ID QT00034566) expression was performed on hdLECs. H2-Ab expression (Qiagen, GeneGlobe ID QT00150332) was performed on FACs-sorted LECs (CD45-Podo+CD31+), DCs (CD45+CD11c+), and macrophages (CD45+CD11b+). RNA isolation was achieved using TRIzol Reagent (Invitrogen, Cat#15596026; Waltham, MA, USA) according to the manufacturer’s instructions. The RNA (1 μg) was reverse transcribed into cDNA using the Maxima H Minus cDNA Synthesis kit (Thermo Fischer Scientific, Cat# M1681). Real-time PCR was performed on a Viia7 PCR system (Invitrogen) with Quantitect SYBR Green reagent (Qiagen, Cat#204143) at 55 °C for annealing temperature and 50 cycles of amplification. All samples were assessed in triplicates.

### 4.6. Flow Cytometry

Tumors were mechanically separated away from peritumoral tissue as previously described for tumoral and peritumoral analysis [42,47]. Skin and tissue directly overlying tumors were used as peritumoral tissue. Tumor-associated tissue, LNs, and ear skin were mechanically disrupted with microscissors, followed by enzymatic digestion. In creating single-cell suspensions of spleens, only mechanical digestion was needed, which consisted of using the back of a 3 mL syringe plunger to homogenize spleens through a 70 μm mesh. Enzymatic digestion of non-splenic tissue was achieved with buffer containing collagenase D, deoxyribonuclease I, and Dispase II (Roche Diagnostics, Cat# 04942078001) incubated at 37 °C for 20–30 min. The enzymatic reaction was stopped using PBS with 2% FCS. Suspensions were filtered with 70 μm followed by 35 μm mesh. Erythrocytes were lysed with RBC lysis buffer (eBioscience; San Diego, CA, USA). Intracellular antibody staining with FoxP3 and Granzyme B was completed with additional sample processing using True NuclearTM mouse Treg flow kit (Biolegend, Cat#320029; San Diego, CA, USA) per the manufacturer’s instructions. For analysis of tumor-infiltrating lymphocytes (TILs), single-cell suspensions were first labeled with CD45 (TIL) MicroBeads (Miltenyi Biotec, Cat#130-110-618), and CD45+ cells were separated using LS columns (Miltenyi Biotech, Cat#130-042-401) and MACS separator (Miltenyi Biotect, Cat#130-090-976).

To determine MHC-II, PD-L1, PD-L2, HVEM, and CD48 expression in primary mouse LECs, single-cell suspensions of peritumoral tissue or LNs were stained with the following antibodies: CD45-FTIC (clone 30-F11, BioLegend, Cat#103108), podoplanin-PE (clone 8.1.1, BioLegend, Cat#127408), CD31-PerCP/Cy5.5 (clone 390, BioLegend, Cat#102419), MHC-II-PeCy7 (clone M5/114.15.2, BioLegend, Cat#107630) or MHC-II-APC (clone M5/114.15.2, Invitrogen, Cat#50-112-9473), PD-L1-APC (clone 10F.9G2, BioLegend, Cat#124312) or PD-L1-PeCy7 (clone MIH7, BioLegend, Cat#155405), PD-L2-BV421 (clone Ty25, BioLegend, Cat#107219), HVEM-APC (clone HMHV-1B18, BioLegend, Cat#136305), CD48-AF700 (clone HM48-1, BioLegend, Cat# 103425). MHC-II and PD-L1 expression in human dLEC cell lines were determined by staining with HLA-DR/DP/DQ-FITC (clone Tu39, BioLegend, Cat# 361705) and PD-L1-PE (clone 29E.2A3, BioLegend, Cat#329705) antibodies. Quantification of splenic DC and macrophages was determined by staining single-cell suspensions with CD45-FITC, CD11c-PeCy7 (clone N418, Invitrogen, Cat#25-0114-82), and CD11b-AF700 (clone M1/70, BioLegend, Cat#101222) antibodies. Quantification of splenic CD8a, CD4, and Tregs were determined by staining single-cell suspensions with CD8a-PerCp/Cy5.5 (clone 53-6.7, BioLegend, Cat#100734), CD4-FITC (clone GK1.5, BioLegend, Cat#100406), CD25-APC (clone PC61, BioLegend, Cat#102012), and FoxP3-PE (clone Mf-14, BioLegend, Cat#126404) antibodies. Analysis of tumor-infiltrating, LN, and splenic CD8+ T cells was performed by staining single-cell suspensions with CD8a-FITC (clone 53-6.7, BioLegend, Cat#100706), granzyme B-PerCp/Cy5.5 (clone QA16A02, BioLegend, Cat#372212), and ovalbumin (OVA) tetramer-PE (NIH tetramer core facility); analysis of CD4+ T cells was performed by staining with CD4-FITC, CD25-APC, FoxP3-PE, and Granzyme B-PerCp/Cy5.5. DAPI viability dye or Zombie UV fixable viability dye (BioLegend, Cat#423108) was used to exclude dead cells. Single-stain compensation samples were created using UltraComp eBeadsTM (Invitrogen, Cat#01-2222). Flow cytometry was performed using a BD Fortessa flow cytometry analyzer (BD Biosciences; San Jose, CA), and data were analyzed using FlowJo software (Tree Star; Ashland, OR, USA).

### 4.7. FACs Cell Sorting

For cell sorting of primary LN LECs, single-cell suspensions were obtained from WT and LEC^MHC-II −/−^ mice by pooling inguinal lymph nodes using the method described above for flow cytometric analysis and stained with CD45-FTIC, podoplanin-PE, CD31-PerCP/Cy5.5. For cell sorting of primary mouse splenic DCs and macrophages, single-cell suspensions were stained using CD45-FITC, CD11c-PerCP/Cy5.5 (clone N418, BioLegend, Cat#117327), and CD11b-AF700 antibodies. Splenic DCs were sorted as CD45+MHCII+CD11c+ and splenic macrophages as CD45+ MHCII+ CD11b+ populations.

Naïve OT-I and OT-II cells were FACs-sorted from pooled spleens and lymph nodes of 6-8-month-old C57BL/6-Tg(TcraTcra)1100Mjb/J (The Jackson Lab. Stock No: 003831) and Tg(TcraTcrb)425Cbn mice (The Jackson Lab, Stock No: 004194), respectively. Sorted OT-I (CD8+CD62+CD44-) and OT-II (CD4+CD62L+CD44-CD25-) cells were cultured in RPMI supplemented with 100 U/mL penicillin, 100 μg/mL streptomycin, 2 mM L-glutamine and 10% FCS in addition to 20 ng/mL rmIL2 (peprotech) or cryopreserved in DMSO for later use. Sorting of naïve OT-I cells were obtained by staining with CD8a-PerCp/Cy5.5 (clone 53-6.7, BioLegend, Cat#100734), CD62L-PeCy7 (clone MEL-14, BioLegend, Cat#104418), CD44-PE (clone IM7, BioLegend, Cat#103007) antibodies; sorting of naïve OT-II cells were obtained by staining with CD4-PerCp/Cy5.5 (clone GK1.5, BioLegend, Cat#100433), CD62L-PeCy7, CD44-PE, and CD25-APC (clone PC61, BioLegend, Cat#102012) antibodies. DAPI viability dye was used to exclude dead cells, and single-strain compensation was created using UltraComp eBeadsTM (Invitrogen, Cat#01-2222). All samples were sorted on a BD FACSAria (BD Biosciences; San Jose, CA, USA). For RNA extraction of FACS sorted cells, cells were sorted directly into trizol buffer (Invitrogen, Cat#15596026) and immediately placed on dry ice until RNA extraction.

### 4.8. Dendritic Cell Migration Assay

Skin DC migration through lymphatic vessels was assessed using a modification of a previously reported method [48]. Briefly, 8% type I isomer FITC (5 mg/mL; Sigma-Aldrich) was diluted in a 1:1 mixture of acetone and dibutyl phthalate (Sigma-Aldrich). A total of 40 μL of the solution was painted on the mouse foot and lower leg 18 h before sacrifice. The inguinal lymph nodes were harvested after sacrifice. Single-cell suspensions were obtained by enzymatic digestion and stained with CD11c-PE (clone N418, BioLegend, Cat#117307) to allow the analysis of FITC+CD11c+ cells per the protocol described above.

### 4.9. BMDC Isolation and Culture

Bone marrow dendritic cells (BMDC) were isolated as previously described [49]. Briefly, the femurs and tibias of euthanized WT mice were removed and placed in cold 70% ethanol for 5 min followed by a 5 min wash in sterile PBS. Then, both ends were cut with the marrow flushed with RPMI using a 0.22 mm syringe into a 70 um mesh. Cells were grown in RPMI supplemented with 100 U/mL penicillin, 100 μg/mL streptomycin, 2 mM L-glutamine and 10% FCS in addition to 20 ng/mL rmGM-CSF (Peprotech, Cat#315-03). On day 3, another 10 mL RPMI containing 20 ng/mL rmGM-CSF was added to the original culture. On days 6 and 8, half of the culture supernatant was collected, and the cell pellet was resuspended in fresh 10 mL RPMI with 20 ng/mL rmGM-CSF. The purity of the culture was checked by flow cytometry at day 6, which was >70% positive for CD11c and MHC-II, at which time cells were used for downstream in vitro experiments.

### 4.10. Tumor Antigen Uptake

B16F10 cells were labeled with CFSE as previously described [50]. Briefly, detached B16F10 tumor cells were resuspended in 1 mL DMEM, stained with 1 μL of 1 μM CFSE (Invitrogen, Cat#34554), and washed twice. Tumor cells were induced to undergo apoptosis through heat-treatment with incubation at 42 °C for 2 h followed by incubation at 37 °C on a shaker at 30 rpm for 24 h. Cell death was confirmed by almost 100% trypan blue staining. The suspension was centrifuged at 1000 rpm × 5 min to remove cell debris. CFSE labeled B16F10 cells were cocultured with the indicated LECs at a 1:1 ratio in 4-well slides (Millipore, Cat#PEZGS0416; Burlington, MA, USA) for 24 h before downstream analysis.

### 4.11. T Cell Proliferation

Day 6 BMDC were matured with 100 ng/mL LPS (Sigma-Aldrich, Cat#L2654) for 24 h before pulsing with 1 mg/mL whole OVA protein (Sigma-Aldrich, Cat#A5503) in RPMI for 24 h in 6-well plates. LECs were concurrently pulsed with 1 mg/mL whole OVA protein in endothelial media with the addition of 5 μg/mL rmIFNγ (Peprotech, Cat#315-05) for 24 h in 6 well plates. For antigen processing inhibition assays, BMDCs and LECs were either pretreated with 20 mM NH_4_Cl (Sigma Aldrich, Cat#A9434) simultaneously as OVA pulsing or pulsed at 4 °C instead of 37 °C for 24 h before co/tri-culture. Pulsed LECs and BMDCs were washed three times the following day before co/tri-culture. OT-I and OT-II cells were labeled with CFSE as described above (tumor antigen uptake) and cocultured with the DCs and/or LECs in 5:1 ratio in 96 round-bottom plates in 200 μL RPMI for 3 days before downstream analysis. For flow cytometric analysis, single-cell suspensions were stained with T-cell receptor (TCR) Va2-AF700 (clones B20.1, BioLegend, Cat#127824) and Granzyme B-PerCp/Cy5.5.

### 4.12. Publicly Available Data

The expression of MHC class I and class II related genes from human breast cancer LECs and normal breast tissue LECs referenced in this study are available in a public repository from the NCBI GEO2R website (https://www.ncbi.nlm.nih.gov/geo/geo2r/) accessed on 30 September 2021 [51].

### 4.13. Statistical Analysis

Statistical analysis was performed using GraphPad Prism (GraphPad Software; San Diego, CA, USA). Unpaired student’s t-test was used to compare differences between two groups, while one- or two-way analysis of variance (ANOVA) was utilized for multiple groups. Data are presented as mean ± standard deviation unless otherwise noted, and *p* < 0.05 was considered significant.

## Figures and Tables

**Figure 1 ijms-23-13470-f001:**
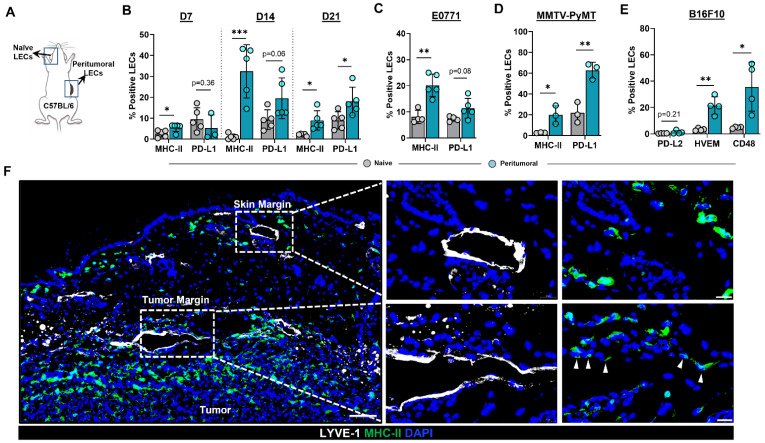
Peritumoral LECs upregulate MHC-II, PD-L1, and various co-inhibitory molecules compared to naïve dermal LECs. (**A**) Schematic of experimental design analyzing naïve dermal LECs in the ear and peritumoral LECs of the same mice. (**B**) Quantification of MHC-II and PD-L1 positivity in naïve and peritumoral LECs (CD45- podoplanin+ CD31+) at day 7, 14, and 21 post B16F10 tumor inoculation in WT (C57/BL6) mice. (**C**) Quantification of MHC-II and PD-L1 positivity in naïve and peritumoral LECs at day 14 post E0771 tumor inoculation in WT mice. (**D**) Quantification of MHC-II and PD-L1 positivity in naïve and peritumoral LECs in MMTV-PyMT mice with spontaneous mammary tumors. (**E**) Quantification of PD-L2, HVEM, and CD48 positivity in naïve and peritumoral LECs at day 21 post B16F10 tumor inoculation in WT mice. (**F**) Representative immunofluorescence image stained for LYVE-1 (white) and MHC-II (green) in peritumoral and skin margin lymphatics in WT mice at day 14 post B16F10 inoculation. DAPI nuclear staining in blue. Scale bars, 100 μm (**left**), 20μm (**top right** and **bottom right**). Statistical significance assessed using paired student’s *t* test on data from at least 3 biological replicates; error represented as SD. *, *p* < 0.05; **, *p* < 0.01; ***, *p* < 0.001.

**Figure 2 ijms-23-13470-f002:**
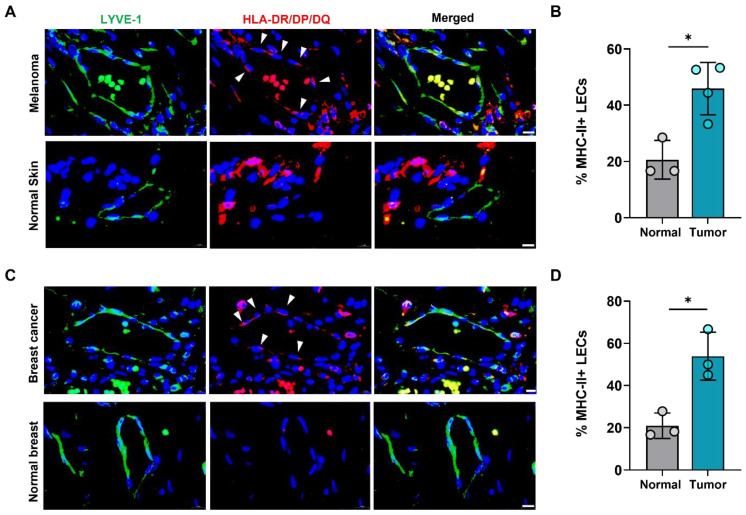
Human peritumoral LECs upregulate MHC-II in melanoma and breast cancer. (**A**) Representative immunofluorescence image and (**B**) quantification of LYVE-1 (green) and HLA-DR/DP/DQ (red) staining in human tissue microarray of melanoma and normal skin tissues. (**C**) Representative immunofluorescence image and (**D**) quantification of LYVE-1 (green) and HLA-DR/DP/DQ (red) staining in human tissue microarray of breast cancer and normal breast tissues. DAPI nuclear staining in blue. Scale bars, 10μm. Statistical significance assessed using paired student’s *t* test on data from at least 3 biological replicates; error represented as SD. *, *p* < 0.05.

**Figure 3 ijms-23-13470-f003:**
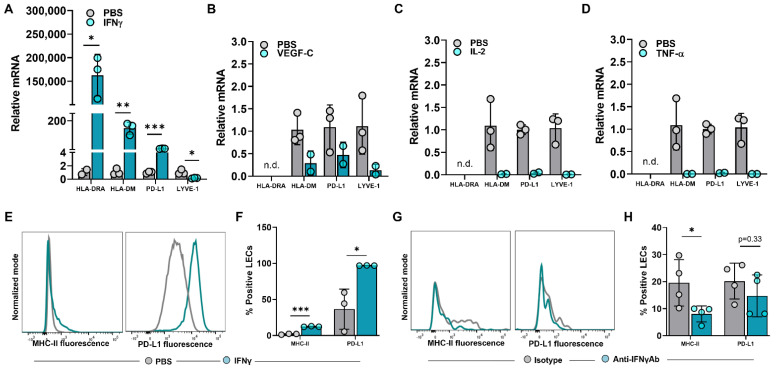
IFNγ induces upregulation of MHC−II in LECs in vitro and in vivo in the tumor periphery. (**A**–**D**) Relative mRNA expression of HLA−DRA, HLA−DM, PD−L1, LYVE−1 in cultured human dermal LECs after 24-h treatment with (**A**) IFNγ (100 ng/mL), (**B**) VEGF−C (100 ng/mL), (**C**) IL−2 (10 ng/mL) and (**D**) TNF−α (10 ng/mL). (**E**) Representative FACS histogram plot and (**F**) quantification of MHC−II and PD−L1 positive human dermal LECs in culture after 24-hour treatment with IFNγ (100 ng/mL) as measured by flow cytometry. (**G**) Representative FACS histogram plots and (**H**) quantification of MHC-II and PD-L1 positive peritumoral LECs in WT mice treated with IFNγ neutralizing antibody or IgG isotype control at day 14 post tumor inoculation. Statistical significance assessed using unpaired student’s *t* test on data from at least 3 biological replicates for cell culture studies and at least 4 biological replicates for mouse experiments; statistical analysis not performed on (**B**–**D**) for which at least 2 biological replicates were used; error represented as SD. *, *p* < 0.05; **, *p* < 0.01; ***, *p* < 0.001. n.d, not detected mRNA in treated group.

**Figure 4 ijms-23-13470-f004:**
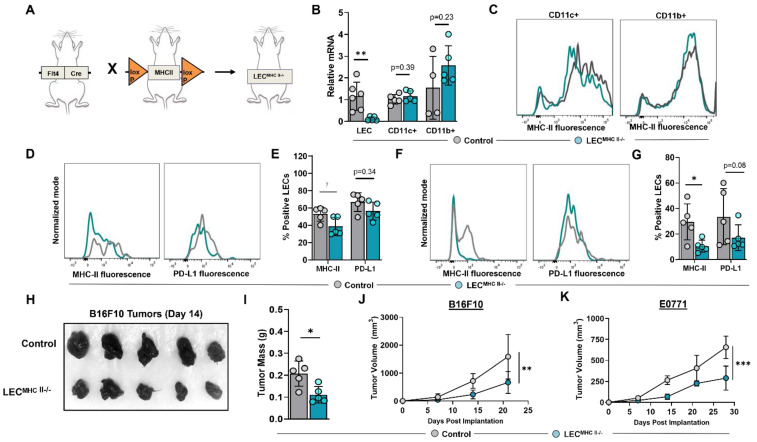
Conditional knock-out of LEC-specific MHC-II expression attenuates growth of heterotopically implanted B16F10 and E0771 tumors. (**A**) Schematic representation of mice genotype bred to generate LEC^MHC II−/−^ mice. (**B**) Relative mRNA expression of MHC-II in FACS-sorted lymph node LECs (CD45-podoplanin+CD31+) and FACS-sorted lymph node CD11c+ and CD11b+ immune cells from WT and LEC^MHC-II−/−^ mice. (**C**) Representative FACS histogram plots of MHC-II MFI in sorted CD11c+ and CD11b+ immune cells from WT and LEC^MHC-II−/−^ mice. (**D**) Representative FACS histogram plots and (**E**) quantification of MHC-II and PD-L1 positive lymph node LECs in WT and LEC^MHC-II−/−^ mice. (**F**) Representative FACS histogram plots and (**G**) quantification of MHC-II and PD-L1 positive peritumoral LECs in WT and LEC^MHC-II−/−^ mice bearing B16F10 at 2 weeks post tumor inoculation. (**H**) Gross images of resected B16F10 tumors and their respective (**I**) mass in grams in WT and LEC^MHC-II−/−^ mice at 2 weeks post tumor inoculation. Statistical significance assessed using unpaired student’s *t* test on data from at least 4 biological replicates; error represented as SD. *, *p* < 0.05; **, *p* < 0.01. (**J**,**K**) Growth curves of B16F10 (**G**) and E0771 (**H**) in WT and LEC^MHC-II−/−^ mice at various time points (in weeks). Statistical significance assessed using two-way ANOVA, error represented as SD.; **, *p* < 0.01; ***, *p* < 0.001.

**Figure 5 ijms-23-13470-f005:**
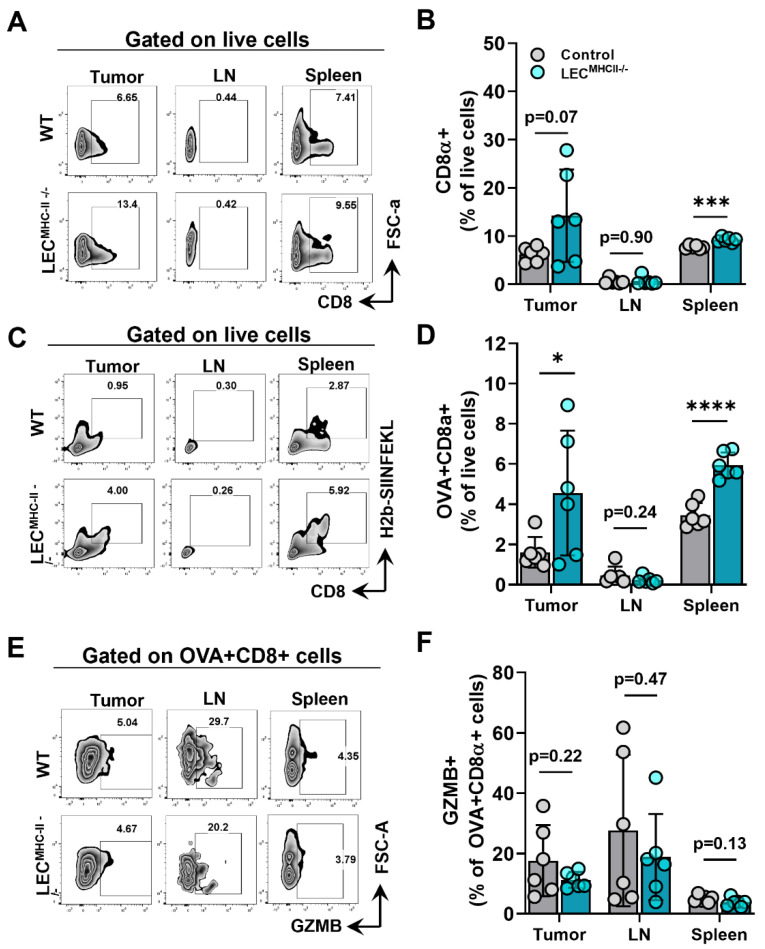
Attenuated tumor growth in LEC-specific MHC-II knockout background is associated with increased number of tumor specific CD8+ TIL. (**A**) Representative FACS plots and (**B**) quantification of CD8α+ cells in the tumor, tumor draining lymph node, and spleen in WT and LEC^MHC II−/−^ mice bearing B16F10-OVA at 12 days post tumor inoculation. (**C**) Representative FACS plots and (**D**) quantification of CD8α+OVA+ cells in the tumor, tumor draining lymph node, and spleen in WT and LEC^MHC II−/−^ mice bearing B16F10-OVA at 12 days post tumor inoculation. (**E**) Representative FACS plots and (**F**) quantification of granzyme B (GZMB)+CD8α+OVA+ cells in the tumor, tumor draining lymph node, and spleen in WT and LEC^MHC II−/−^ mice bearing B16F10-OVA at 12 days post tumor inoculation. Statistical significance assessed using unpaired student’s *t* test on data from at least 5 biological replicates; error represented as SD. *, *p* <0.05; ***, *p* < 0.001; ****, *p* < 0.0001.

**Figure 6 ijms-23-13470-f006:**
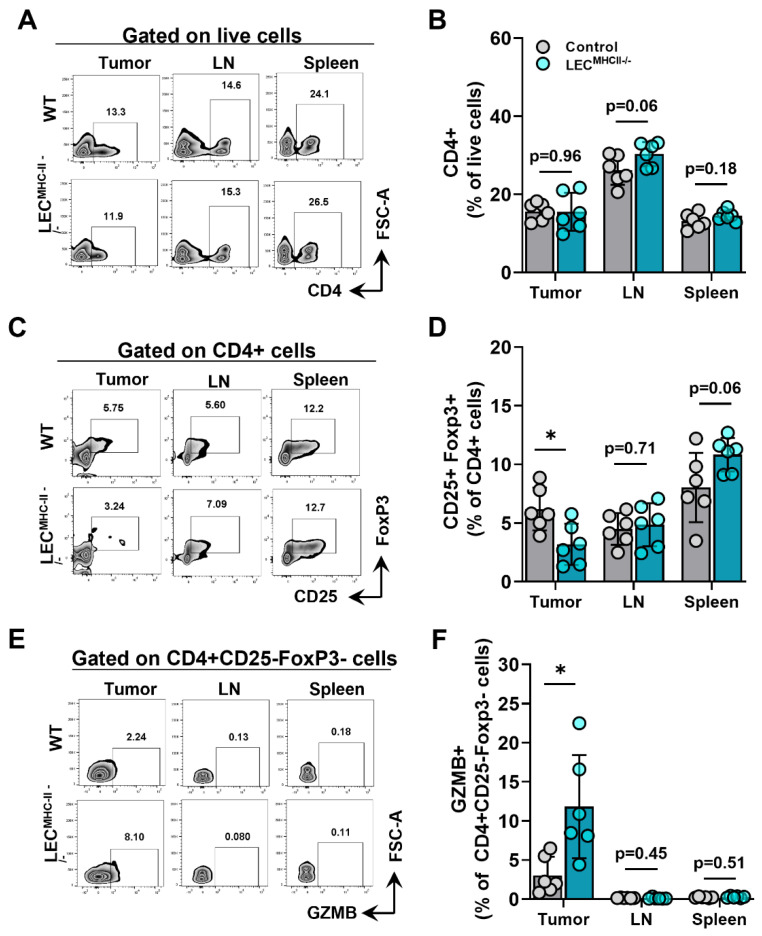
Attenuated tumor growth in LEC-specific MHC-II knockout background is associated with increased number of effector CD4+ TILs and decreased number of T regulatory lymphocytes. (**A**) Representative FACS plots and (**B**) quantification of CD4+cells in the tumor, tumor draining lymph node, and spleen in WT and LEC^MHC II−/−^ mice bearing B16F10-OVA at 12 days post tumor inoculation. (**C**) Representative FACS plots and (**D**) quantification of CD4+CD25+FoxP3+cells in the tumor, tumor draining lymph node, and spleen in WT and LEC^MHC II−/−^ mice bearing B16F10-OVA at 12 days post tumor inoculation. (**E**) Representative FACS plots and (**F**) quantification of GZMB+CD4+CD25-FoxP3- in the tumor, tumor draining lymph node, and spleen in WT and LEC^MHC II−/−^ mice bearing B16F10-OVA at 12 days post tumor inoculation. Statistical significance assessed using unpaired student’s *t* test on data from at least 5 biological replicates; error represented as SD. *, *p* < 0.05.

**Figure 7 ijms-23-13470-f007:**
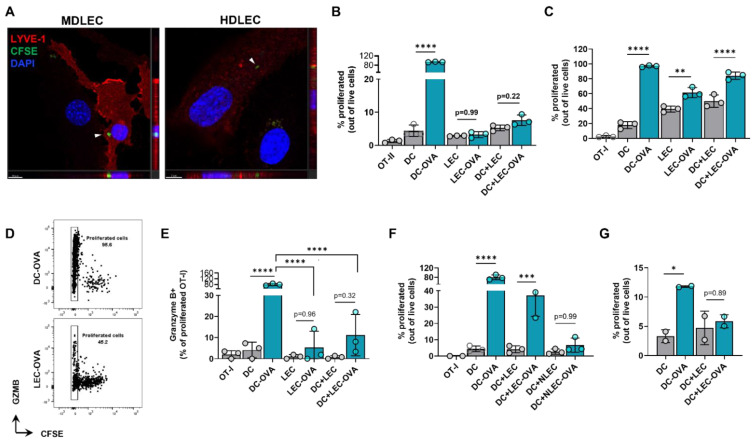
Cultured LECs engulf tumor debris and induce CD8+ T cell proliferation with reduced effector function. (**A**) Representative confocal micrograph including Z-plane of CFSE-labeled B16F10 tumor debris (arrows) within cultured mouse dermal LECs (MDLEC) and human dermal LECs (HDLEC). Scale bars; left, 10 μm; right, 7 μm. (**B**) CD4+ OT-II and (**C**) CD8+ OT-I proliferation after co/tri-culture conditions with the indicated cell types in the presence of LEC or DC unpulsed or pulsed with OVA protein (2 mg/mL) 24 h prior to co/tri-culture. (**D**) Representative FACS plot and (**E**) quantification of GZMb+ proliferated OT-I cells after culture with indicated cell types in the presence of LEC or DC either unpulsed or pulsed with OVA protein (2 mg/mL) 24 h prior to co/tri-culture. (**F**) OT-I proliferation after co/tri-culture with indicated cell types either NH_4_-Cl pretreated or not for 24 h followed by pulsing or no pulsing with OVA protein (2 mg/mL) 24 h prior to co/tri-culture. (**G**) OT-I proliferation after culture with indicated cells pulsed or not pulsed with OVA protein (2 mg/mL) at 4 °C 24 h prior co/tri-culture. Statistical significance assessed using one-way ANOVA on data from at least 3 biological replicates (B-G); DC-OVA, OVA pulsed DC; LEC-OVA, OVA pulsed LEC; NLEC, NH_4_-Cl pretreated LEC; NLEC-OVA, OVA pulsed NH_4_-Cl pretreated LEC..Error represented as SD. *, *p* < 0.05; **, *p* < 0.01; ***, *p* < 0.001; ****, *p* < 0.0001.

## Data Availability

Use of publicly available data is detailed in the methods section.

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
