# Peer review of "Tumor-Associated Lymphatics Upregulate MHC-II to Suppress Tumor-Infiltrating Lymphocytes"

_ijms, 2022, doi:10.3390/ijms232113470_

Round 1

Reviewer 1 Report

The study by Li et al is focused on unravelling the mechanisms underlying the immune-modulatory potential of tumor lymphatics. Recent investigations have revealed that lymphatic vessels in peripheral tissues may act as antigen-presenting cells (APCs) in pathology. As lymphatics are enriched in co-inhibitory signals, rather than co-stimulating signals, this lymphatic-lymphocyte cross-presentation leads to T cell anergy and antigen tolerance. In the settings of cancer this may lead to tumor-immune escape. However, it remains to be better described the molecular pathways underlying the functional impact on T cell anti-tumor immune responses and conversely how the tumor microenvironment alters lymphatic vessel immune repertoire. Specifically, the authors hypothesized that tumor lymphatic vessels may cross-present tumor antigens not only via MHC-I but also via MHC-II molecules, which would allow lymphatics to interact directly with both CD8 and CD4 T cells.

Elegantly investigating several murine tumor types, both ectopic and orthotopic, as well as two human cancer types, the authors demonstrate consistent upregulation of MHCII in peritumoral lymphatics. Further, both murine and human tumor lymphatics were found to display increased co-inhibitory molecule expression, including both PDL1 and HVEM in mouse models, and only HVEM in human tumors. Next, they set out to characterize what cytokine(s) in the tumor microenvironment drive this APC activity and co-inhibitory signal expression in human lymphatic endothelial cells (LEC). They demonstrate that IFNγ, but not IL2 or TNFα, increase lymphatic MHCII and PDL1 expression in vitro. Conversely, anti- IFNγ treatment in vivo reduced tumor lymphatic MHCII expression. Interestingly, a key lymphangiogenic growth factor VEGF-C, often enriched in the tumor microenvironment, tended to reduce the levels of MHCII catalyst HLA-DM, which would oppose the stimulatory effects of IFNγ on MHCII presentation in LECs.

To test the functional relevance of MHCII expression in tumor lymphatics, the authors then developed a conditional mouse model lacking MHCII expression in VEGFR3 (Flt4)-expressing lymphatics. While this mouse model did not display altered immune phenotype in physiology, following tumor implantation the splenic, but not tumoral, levels of total CD8 T cells was increased. In agreement with improved anti-tumor immune responses, tumor growth was reduced in mice lacking lymphatic MHCII expression. Next, using ovalbumin (OVA)-expressing tumors to model tumor-specific antigen-processing by lymphatics, the authors demonstrate reduced levels in the tumor microenvironment of CD4 Tregs and increased activity of CD4 effector T cells coupled to increased numbers of OVA-reactive CD8 T cells in Flt4-MHCIIiLECKO knockout mice. These data strongly support their hypothesis of immune-suppressive functions of tumor lymphatics being at least partly exerted via MHCII cross-presentation mechanisms. Finally, the authors examined the potential direct tolerogenic effects of LEC antigen presentation on OT1 vs OT2 T cells in vitro. Interestingly, they show that while OVA-pulsed LECs failed to induce OT1 naïve CD4 T cell proliferation, it increased OT2 naïve CD8 T cell proliferation, albeit less potently than Ova-presenting dendritic cells. However, LEC presentation of Ova did not suffice to stimulate CD8 T cell activity, indicating reduced effector function, in line with direct tolerogenic effects.

Comments:

1)      Previous studies have suggested that LECs induce CD8 memory phenotype, do the authors have any information on whether CD8 memory population in altered in mice with MHCII-deficient lymphatics?

2)      Fig. 1E vs Suppl Fig 2F: please comment on the inversion of relative MHCII vs PDL1 expression noted in tumor lymphatics

3)      Suppl Fig 2B, D, F: please check axis legend (currently stated as % but looks more like cell counts)

4)      The mention of CD48 as a strictly coinhibitory molecule seems overly simplified, as it has been shown to stimulate IL2 production in naïve CD2-expressing CD8 T cells (Latchman et al, Journal of immunology. 1998). In contrast, it may indeed have co-inhibitory effects in NK cells and in a subset of CD8 T cells, ie CD244-expressing memory T cells. Is there any data that peritumoral CD8 T cells in the authors’ tumor models preferentially express CD244 rather than CD2?

5)      Fig. 3 B, C, D: why were expression levels of HLA-DR excluded from these graphs?

6)      Fig. 3C, D: Did the authors assess consider whether IL2 or TNFa exposure was cytotoxic to LECs?

7)      Suppl. Fig. 3C: legend (in the dotplot) indicates “monocytes” but the population shown is macrophages

8)      Please add quantification of Cd11b+ F4/80- myeloid cell (monocyte or DC) MHCII expression levels in LECMHCII-/- mice

9)     Fig. 7 F: Please indicate the groups in the legend (N=ammoniumchloride-treated cells?).

Reviewer 2 Report

Li and colleagues investigated lymphatic endothelial cell (LEC) mediated regulation of immune cell responses in the tumor microenvironment employing in vivo mouse models and human and mouse in vitro culture systems with appropriate quantification approaches followed by statistical analyses. That EC can present antigen in a MHC restricted way is an area of controversy in vascular cell biology and immunology.

The main findings reported were that MHC-II expression in peri-tumor lymphatic endothelium (LEC) was elevated in three mouse tumor models and also in human breast and melanoma tumor tissue microarrays. In vivo and in vitro culture studies showed stimulation of LEC by IFN-g, but not TNF-a, IL-2 or VEGF-C, induced MHC-II and HLA-DM expression, and upregulated co-inhibitory molecules PD-L1and CD48. Murine melanoma and breast tumors implanted in mice with conditional excision of MHC-II in LEC had reduced tumor mass, and no alteration in lymphatics in tumor area. Subsequent studies were performed to dissect the mechanisms for attenuated tumor growth report. Authors conclude that loss of MHC-II specifically in peritumor LEC leads to less immunosuppressive pressures via alterations in tumor infiltrating lymphocytes, including reduction in CD4+ T-regs and an increase in CD8+ T cells with modest levels of Granzyme B.The manuscript is well written and has good controls and data presentation. The results are of interest and confirm a recent publication, but certain of the key data requires further supporting evidence regarding MHC-II expression and excision.

Weakness: Clarifications and additional data and/or revising of data are needed to provide a more convincing report.

1.    Results Fig 1B – D, F and Sup F1 and 2. Elevated MHCII, PD-L1, MVEM and CD48 expression in LEC in peri-tumor areas needs to be more convincingly demonstrated. Suggest in Figure 1 and SFig1 include the FACS staining scheme w/ histograms identifying CD31+Podoplanin+, CD45- LEC cells and their levels of CD48, MVEM, MHC-II and PD-L1, in addition to existing bar plots. SFig1B demonstrates  there are very few (in number) LEC that comprise the peritumor tissue. The immunofluorescence  image alone in Fig 1 panel F is weak evidence. Authors should quantify by measuring a number of "positive" vessels (% positive) in peri-tumor/tumor/skin margin regions of replicates and state the method/approach used determine a "positive" vs background signal 

These conclusions are important and main findings, thus require more robust supporting data. For example, the increase in PD-L1 is minimal by flow cytometry (SFig 2) and a small % of the total LEC population, except for the MMTV-PyMT data set.  Solidifying the expression of these molecules is important since Authors do not provide direct evidence that MVEM, CD48 or PD-L1 are physiologically functional.

2.    Results: Extent of tamoxifen induced excision of MHC-II in LEC of Tg mice is unclear/never stated as the mRNA seems misleading given the comment in the Discussion (see below). Suggest Authors state the amount of excision in Results (use a bar plot) and better demonstrate the extent of MHC-II excision in LEC of conditional KO mice treated with tamoxifen vs non-tamox treated Tg mice. Authors further cloud this situation by stating that excision is not robust (Discussion, lines 392-395). This lack of clarity raises the question of the source of MHC-II in the model. Could MHC-II have been derived and transferred from peritumor DCs, potentially diminishing the role of LEC MHC-II in this story?   Specifically, in SFig 3, the arrow points to blood EC, not LEC. Fig 4B gives only mRNA, but could be improved by including the mRNA plot with representative histograms of MHC-II staining of LEC from conditional MHC-II vs non-tamoxifen treated Tg mice. Here it is important that the control are littermates Tg mice that were not treated with tamoxifen.

3.    Figure 1C: Were additional co-Inhibitory molecules are upregulated in LEC? Are there other co-inhibitory potentially involved in suppressing immune cells activity given the relatively low levels of PD-L1?

4.    Fig 3E, I may have missed it, so can Authors include representative FACS histograms for PD-L1 and MHCII; if data were in supplemental, move to fig 3E. Was HLA-DRA expression detected/altered in LEC treated with VEGF-C, IL-2 or TNF? If not, assessed should be stated in figure and legend.

5.    Fig 5A. The tumor FACS plot in 5A was not representative of data’s mean value in panel B. Suggest authors select another plot that better matches the mean value of bar plot.

Minor:

·      Discussion, line 422; think you meant to use the noun “we” instead “was”

·      Y-axis in Sup Fig 2 plots B,D,F appear to be fluorescent intensity, not % Positive LEC. Clarify label.

·      Lines 276-7 think you meant decrease in the # of CD4 FoxP3+ cells in KO

Reviewer 3 Report

In the manuscript “Tumor-associated lymphatics upregulate MHC-II to suppress tumor-infiltrating lymphocytes”, authors preciously inspected expression of MHC-II on lymphatic endothelial cells (LEC) in the tumor and found that its upregulation. Although similar report has been published by Gkountidi et al. in mouse study, this manuscript not only recapitulate it but also broaden it to human specimen. Unfortunately, this study could not present direct evidence of induction of regulatory T cells by LECs in vitro, LEC-specific knockout of MHC-II speculates regulatory T cell-mediated immunosuppression caused by intratumor LECs. The manuscript is written well and could be accepted for publication in International Journal of Molecular Science with minor revisions.

Major comments:

1. line 109, authors argue that upregulation of MHC-II is not induction of expression in a subset of cells by histogram of fluorescent intensity. However, Figure 2S A shows a two-peak pattern and lower peak accords with staining of ear skin LEC suggesting there is at least two subpopulation that upregulates MHC-II and not.

2. Figure 4C, the extent of MHC-II KO is inspected by FACS analysis and shown in % of positive cells. This plot does not have information of MHC-II expression levels. It would be preferable to be presented in histogram of MHC-II FACS or mean fluorescent intensity (MFI).

3. line 396, the authors hypothesize that MHC-II should be depleted after tumor implantation to get clearer result. I do not understand authors’ argument because MHC-II KO before tumor implantation should negate TME-dependent MHC-II upregulation similarly.

Minor comments:

1. line 133, description “tumor microarrays” is not clear. Tissue microarray instead?

2. line 411, “LEC MHC-II-/- was associated with an increased Treg suppressive phenotype” is not clear. Do authors mean “LEC MHC-II was associated with…”?

3. line 422, “The fact that was did not see CD4+ T cell” Is this mistyping of “The fact that we did not see…”?

Round 2

Reviewer 2 Report

In the revised ms by Li and colleagues, the Authors have done a good job of addressing my concerns by adding additional data and revising the text. There are, however, a few but necessary issues on data that need to be addressed.

1.    In bar graphs of Figure 2B and 2D, there are three bars that have only 2 data points, and Fig 3 control bars appear to have less than 3. Statistical analyses on two data points, as is done here, is not acceptable. The minimum data replicates/exp/samples are three or more data points. Authors need to proof and verify in remaining figures that this same issue has not occurred.

2.    The Authors labeling of supplemental figs in text as “slides”  is totally confusing and not acceptable for readers. Data referred to in the text are not actually in that slide!! Slide data should be incorporated into sup Figures, not the other way around as done here.

Round 3

Reviewer 2 Report

Authors have corrected the statistical analytics in figs/text and revised text for Sup Figs  as suggested.